# Using Ex Situ Seedling Baiting to Capture Seedling-Associated Mycorrhizal Fungi in Medicinal Orchid *Dendrobium officinale*

**DOI:** 10.3390/jof8101036

**Published:** 2022-09-29

**Authors:** Yi-Hua Wu, De-Yun Chen, Xin-Ju Wang, Neng-Qi Li, Jiang-Yun Gao

**Affiliations:** Institute of Biodiversity, School of Ecology and Environmental Science, Yunnan University, Kunming 650500, China

**Keywords:** endophytic fungi, orchid conservation, orchid mycorrhizal fungi, ex situ baiting, seedling-associated fungi

## Abstract

Using orchid mycorrhizal fungi (OMFs) to facilitate orchid proliferation is considered an effective method of orchid conservation. Based on the success of using in situ seedling baiting to obtain plant growth-promoting fungi in our previous study, in this study, we developed the method of using ex situ seedling baiting to capture seedling-associated fungi from *Dendrobium officinale*. We collected substrates (e.g., litters, barks and mosses) from six original habitats of *D. officinale* in different geographical locations in China, and then, transplanted in vitro-produced seedlings of *D. officinale* into the substrates. After cultivation for 75 days, it was obvious that fungi colonized the seedling roots and formed large numbers of pelotons in all six groups. From these seedling roots, a total of 251 fungal strains, which were divided into 16 OMF and 11 non-OMF species, were successfully isolated. The 16 OMFs included 13 *Tulasnella* and 3 Serendipitaceae species. The fungal species isolated from the different groups (original habitat sources) were not identical, but the dominant OMFs with high isolation frequencies (more than 10 times) were commonly isolated from more than four original sources. Among the 11 non-OMFs, *Fusarium oxysporum* TP-18 and *Muscodor* sp. TP-26 were the dominant endophytes. *Fusarium oxysporum* is a common endophyte associated with many orchid species, including *D. officinale*. The results suggest that ex situ seedling baiting is an easy and efficient approach to obtaining seedling-associated fungi for this species and could be performed for other over-collected species, especially orchids for which wild plants have disappeared in the field but their original habitats are known. This approach has great potential for application in OMF studies in the future.

## 1. Introduction

Mycorrhiza is a symbiotic association between plant roots and fungi, and is commonly found in most land plants [1]. Based on fungus interactions with the host plant root, mainly the structures of fungal hyphae that form a symbiotic interface with the host cells, four distinct types of mycorrhizal associations are currently recognized: arbuscular mycorrhiza (AM), ectomycorrhiza (EcM), ericoid mycorrhiza (ErM) and orchid mycorrhiza (OM) [2]. Different mycorrhizal associations have distinct evolutionary histories, anatomies and ecologies, thereby differentially affecting plant protection, nutrient acquisition and cycling [3,4]. Mycorrhizal fungi (MFs) mostly benefit plants by enhancing their nutrient access and stress tolerance, and therefore, strongly affect plant populations and community biology by regulating seedling establishment and species coexistence [4]. This symbiosis is particularly important for orchids because most orchids completely rely on mycorrhizal fungi for seed germination [3,5].

Orchids show complex symbioses with fungi during their lifespan, and orchid mycorrhizal fungi (OMFs) may change at different plant developmental stages [6,7,8]. Many studies suggest that OMFs could influence local orchid abundance and population dynamics because the broadening and/or changing of a mycorrhizal association enables orchids to adapt to varied physiological changes during seedling development [9,10]. Orchidaceae is one of the most species-rich families of flowering plants, but these are also among the most threatened flowering plants. In the conservation of orchids, one of the biggest challenges is posed by their complex life history strategies; in particular, orchids require mycorrhizal fungi for seed germination and use specific animals for pollination [11,12]. Using OMFs has been demonstrated to improve the success of seed-based orchid conservation, and is considered an effective method of orchid conservation [13,14,15].

In most studies, the OMFs used for symbiotic germination or plant growth are obtained from the roots of wild adult plants [16,17,18]. However, this approach is a hard task and might not always obtain samples from the optimal source of fungi for seed germination or plant growth [19]; this is because a high diversity of fungi may associate with adult plant roots, and orchids may associate with different mycorrhizal fungi at different life stages [11,20,21]. With increasing studies, in situ/ex situ seed baiting, in which fungi are obtained from naturally formed protocorms or seedlings, is demonstrated as an effective method to obtain efficient OMFs for seed germination [22,23]. In our previous studies, we used in situ/ex situ seed baiting to successfully obtain efficient germination-promoting fungi for different orchid species [24,25,26,27,28,29], and used the obtained fungi successfully in the practices of orchid conservation [29,30]. Considering the fungal dynamics at the orchid seedling stage and whether the fungi associated with seedlings remain until plants reach the adult stage, to obtain plant growth-promoting fungi, the most straightforward method is to isolate fungi from naturally growing seedlings. However, sampling seedling materials and identifying the species of seedlings in the field are very difficult, and for those orchids that have been heavily over-collected and are on the verge of extinction in the wild, e.g., *Dendrobium officinale* [31], it is almost impossible to obtain seedling materials or even adult plants. In our previous studies, by transplanting in vitro-produced seedlings of *D. officinale* into their original habitats, we used newly established roots of well-growing seedlings to isolate fungi after more than one year, and successfully obtained five *Tulasnella* species, including one isolate that was able to strongly promote seedling growth [19]. The fungus has not been reported among OMFs isolated from protocorms or adult roots of *D. officinale* in previous studies, suggesting that symbiont switching may occur depending on the life stage of *D. officinale*, and that seedlings of *D. officinale* established new fungal partners that, in turn, played an important role in their growth.

*Dendrobium officinale* is one of the most important orchid species used in traditional Chinese medicine, with a long history in China. Although it is widely distributed in subtropical areas of China, it has been over-collected to the point of local extirpation [31]. It is included as critically endangered on the IUCN Red List [32] and listed under Class II on the Chinese National Key Protected Wild Plants List (http://www.forestry.gov.cn/main/5461/20210908/162515850572900.html, accessed on 10 September 2021). Therefore, it has received much research attention regarding its mycorrhizal symbionts, including the isolation and identification of OMFs for seed germination and plant growth [33], making it an ideal species for studies of fungal diversity among different habitats, as well as different plant developmental stages.

Motivated by the success of the previous studies, we decided to use this well-studied medicinal orchid, *D. officinale*, to carry out ex situ seedling baiting, an approach expected to be simpler and more efficient, to capture and obtain seedling-associated OMFs for further screening of seedling growth-promoting fungi in *D. officinale.* Here, we present our results, aiming to test: (a) if this approach could obtain seedling-associated fungi in *D. officinale,* and whether is more efficient and easier than the method of in situ seedling baiting that we used in a previous study; (b) the differences in dominant fungi indicated by isolation frequencies among different habitats/original sources; and (c) the phylogenetic relationships among the OMFs obtained in the current study and those obtained from the protocorms or adult plant roots of *Dendrobium* species in previous studies.

## 2. Materials and Methods

### 2.1. Study Species

*Dendrobium officinale* Kimura et Migo is a small epiphytic orchid. It was recorded as being widely distributed in subtropical areas at altitudes from 500 to 1600 m in China. It grows on tree trunks in forests, on rocks in karst landforms or on sandy conglomerates in the Danxia landform [34].

### 2.2. Sample Collection and Ex Situ Seedling Baiting Experiments

Based on the literature and specimen records, we developed a detailed field survey plan to visit the sites where *D. officinale* is most likely to be distributed and is known to have grown for several years. During August 2018, we conducted field trips to Yunnan, Hunan, Guizhou, Sichuan and Chongqing, which cover most distribution areas of *D. officinale* in China. At each site, we found a local medicinal plant harvester or an orchid cultivator to guide us in visiting the specific location where plants of *D. officinale* had been collected or found not long ago. Finally, six sites were confirmed as original habitats of *D. officinale,* as we found and identified some naturally growing seedlings of *D. officinale* there. At each of the six sites, mosses and litters on rocks, barks and mosses on trees, or mosses and topsoil on sandy conglomerates were sampled according to where the plants of *D. officinale* grew (Table 1). At each site, available materials within 30 cm around the plants were randomly collected up to about 5 kg, put together as original substrates and transferred to the laboratory within 48 h. Then, the collected materials from each site were mixed together with a sterilized mixed substrate (bark, peat and volcanic stone mixed in a ratio of 2:1:1 in volume and autoclaved at 12 °C for 30 min) in equal volumes, and then, potted and irrigated with sterile water until saturation.

Seeds of *D. officinale* were obtained via hand pollination between different cultivated individuals, and then, asymbiotically germinated on MS medium [35] for about 8 months to obtain in vitro-produced seedlings with 5–6 leaves for the ex situ seedling baiting experiment. Five seedlings were transplanted into each circular plastic pot (height × diameter = 8 cm × 10 cm) with the mixed materials (Figure 1a). Finally, 6 treatments/groups corresponding to the 6 sampling sites were implemented, and each treatment was replicated across 60 pots. To avoid possible fungal infection among groups, all pots in a given group were placed in a separate incubator under 75 ± 5% relative humidity at 25 ± 2 °C with a 12/12-h light/dark cycle.

### 2.3. Fungal Isolation and Identification

To determine the time point of root sampling for fungus isolation, three roots from each group were randomly sampled and stained to examine them for the presence of pelotons at 30, 45, 60, 75 and 90 days after cultivation, respectively. The roots were first treated with 10% KOH solution, bleached in 0.5% H_2_O_2_, neutralized in 1% HCl solution, then stained in 0.05% (*w*/*v*) trypan blue of glycerol acetate solution for 30 min at 37 °C and decolored with glycerol acetate for 24 h; then, they were observed under the microscope for the formation of pelotons. The sampled roots were also embedded in LR white resin, and semithin sections were stained with 1% (*w*/*v*) toluidine blue to observe and record the pelotons inside the cells. Once large numbers of pelotons were observed in the roots, indicating that mycorrhizal symbiosis was actually established, we began to sample the roots for fungal isolation.

For each of the 6 treatments, at least 40 roots were randomly collected. The roots were surface-sterilized with 75% alcohol for 2 min and 1% (*w*/*v*) NaClO for 3 min, and then, washed with sterile distilled water 3–5 times. After the superficial uncolonized tissues were removed, root fragments containing intracellular pelotons were scraped to be as small as possible using an anatomical needle under a 10 × 20 magnification microscope; then, they were transferred to PDA medium (200 g/L potato, 20 g/L dextrose and 20 g/L agar; pH = 5.6) with 0.05 g/L penicillin and 0.05 g/L streptomycin and incubated at 25 ± 2 °C. When hyphal growth from the root fragments exceeded 0.5 cm in length, the tips of the hyphae were cut and transferred to new PDA medium for purification. After repeating this purification step 4–5 times, purified strains were obtained [19].

Among all the obtained fungal strains, those fungal strains in which the colony colors presented as pink, green and black were first discarded, and only white and transparent strains were used for morphological studies on colony morphology and hyphal characteristics [36]. Then, all the fungal strains were used for molecular identification except for those strains isolated from the same root fragments with high morphological similarity, in which one fungal strain was randomly selected for molecular identification. After DNA extraction using a Fungi Genomic DNA Extraction Kit (D2300, Solarbio, Beijing Solarbio Science & Technology Co.,Ltd., Beijing, China), the rDNA region containing the two ITS regions and the 5.8S rRNA gene was amplified using ITS1 and ITS4 primers [37]. The PCR reaction consisted of the following steps: initial denaturation at 94 °C for 3 min, denaturation at 94 °C for 1 min, annealing at 51 °C for 1 min, 30 cycles of extension at 72 °C for 1 min, and final extension at 72 °C for 10 min. All the amplification products were purified and sequenced bi-directionally at Shanghai Personal Biotechnology Co., LTD., Shanghai, China [24]. All the ITS sequences obtained were used in homology searches using BLAST against the GenBank database (National Center for Biotechnology Information, Bethesda, MD, USA)—which allows the identification of isolates to the genus or species level when ITS sequence similarity exceeds 95% or 99%, respectively [38]—and were deposited in GenBank with the accession numbers MN918475–MN918501 (Table 2).

### 2.4. Phylogenetic Analysis

We collected the available literature about mycorrhizal fungal studies of *Dendrobium* species, and the OMFs originally isolated from *Dendrobium* species with information on original their geographical occurrences and sources were selected for comparison with the OMFs obtained in the current study (Appendix A). Then, the available ITS rDNA sequences of the OMFs were downloaded from the GenBank database (https://www.ncbi.nlm.nih.gov/, accessed on 15 April 2021), and phylogenetic trees were generated for different fungus taxa to determine the phylogenetic relationships of the OMFs obtained in the current study with other known OMFs associated with *Dendrobium* species.

All the sequences were first aligned using MAFFT software, and then, manually adjusted using BioEdit software (v7.0.9) [39]. Maximum-likelihood (ML) phylogenetic tree searches and ML bootstrapping were conducted using the web server RAxML-HPC2 on TG ver. 7.2.8 with 1000 rapid bootstrap analyses, followed by a search for the best-scoring tree in a single run [40]. All the parameters in the ML analysis were kept at their default levels, and 1000 bootstrap replicates were applied to obtain a statistical support. The sequences of *Ceratobasidium* sp. (KF176584) and *Dacrymyces* sp. (AY463403) served as outgroup sequences of Serendipitaceae and Tulasnellaceae, respectively. The phylogenetic trees were uploaded to the iTOL website (www.itol.embl.de, accessed on 16 May 2021) for visual improvement.

## 3. Results

### 3.1. Fungal Isolation and Identification

In the ex situ seedling baiting experiments, at 75 days after cultivation, large numbers of pelotons were observed in the roots of *D. officinale* plants in all six treatments (Figure 1b). A total of 320 root segments were randomly sampled in 10 runs, and 600 root fragments were used for fungal isolation. A total of 251 purified fungal strains were successfully obtained. Based on the morphological studies and whether fungal strains were isolated from the same root fragments, 236 out of 251 purified fungal strains were finally used for molecular identification. Then, the 236 strains were sequenced and molecularly identified as 27 fungal species, including 16 OMFs (TP-1 to TP-16) and 11 non-OMFs (TP-17 to TP-27) (Figure 2; Table 2).

The 16 OMFs included 13 species of *Tulasnella* and 3 species belonging to Serendipitaceae. Among them, *Tulasnella* TP-13 was isolated from all six sites, with the highest isolation frequency being 20 times. Four fungi, namely, *Tulasnella* TP-9, *Tulasnella* TP-2, *Tulasnella* TP-11 and Serendipitaceae TP-15, were obtained from five original sources and isolated 19, 14, 11 and 9 times, respectively. *Tulasnella* TP-7 and TP-8 were isolated from four original sources with isolation frequencies of 6 and 15 times, respectively. Three fungi, namely, *Tulasnella* TP-1, TP-3 and TP-10, were isolated five, eight and eight times from three original sources, respectively. Five fungi were isolated from two original sources with isolation frequencies of 2–7 times. *Tulasnella* TP-6 was isolated only from the Danxia landform of Langshan Mountain in Hunan (DXLS), with an isolation frequency of four times (Table 2). In terms of the sampling sites, 13 out of 16 OMFs were obtained from the LT site, with the highest isolation frequency among the six sites being 31 times (Table 2).

Among the 11 non-OMFs, *Fusarium oxysporum* TP-18 was isolated from four original sources but had the highest isolation frequency of 27 times, and *Muscodor* sp. TP-26 was isolated from all six original sources, with an isolation frequency of 24 times. Two *Ple**ctosphaerella* fungi, *P. niemeijerarum* TP-20 and *P. cucumerina* TP-21 samples were isolated from four original sources 10 and 13 times, respectively, while the other seven non-OMFs were obtained at low frequencies (fewer than 7 times) (Table 2).

### 3.2. Phylogenetic Analysis

We found 19 studies documenting 74 fungi isolated from *Dendrobium* species. Among them, 38 OMFs were from *D. officinale* with different original sources covering most of the distribution areas of the species in China, including Guizhou, Hunan, Sichuan, Guangxi, Yunnan, Zhejiang and Chongqing (Appendix A). For the 38 OMFs from *D. officinale*, 16 fungal species/strains were isolated from the roots of adult plants, another 16 fungal species/strains were isolated from the protocorms/germinated seeds, 5 were isolated from seedling roots, and only Sebacinales LQ (MN173026) was obtained from both adult roots and protocorms (Appendix A). The ITS-rDNA sequences of 61 of the 74 OMFs were downloaded from the GenBank database. Combining these OMFs with the 16 OMFs obtained in the current study yielded a total of 77 OMFs, including 17 species of Sebacinales and 60 species of Tulasnellaceae, which were used to generate two phylogenetic trees of Sebacinales and Tulasnellaceae, respectively (Figure 3).

In the ML tree of Sebacinales, the 17 OMFs fell into two clades. The three species of Serendipitaceae, TP-14, TP-15 and TP-16, were clustered in Clade II with another nine species/strains, while five species/strains of Sebacinaceae, including two isolated from protocorms of *D. chrysanthum* and three from the roots of *D. nobile*, were clustered in Clade I (Figure 3a). For the ML tree of Tulasnellaceae, all 60 OMFs fell into five clades (Figure 3b). The seven species/strains of *Tulasenlla*, all isolated from protocorms of *D. aphyllum*, were independently grouped as Clade I. For the thirteen *Tulasnella* species obtained in the current study, three *Tulasnella* species (TP-6, TP-8 and TP-13) with sixteen species/strains of Tulasnellaceae and three strains of *Tulasnella calospora* were clustered in Clade II; seven *Tulasnella* species (TP-2, TP-5, TP-7, TP-9, TP-10,TP-11 and TP-12) were clustered in Clade III; with five species/strains of *Tulasnella* obtained from another five studies; and for the other three fungi (TP-1, TP-3 and TP-4) with relatively low isolation frequencies, TP-3 and TP-4 were clustered in Clade IV with nine fungi/strains from other studies, and TP-1 with seven fungi/strains from other studies were clustered in Clade V (Figure 3b).

## 4. Discussion

### 4.1. Using Ex Situ Seedling Baiting to Obtain OMFs Associated with Seedlings

OMFs commonly belong to the so-called rhizoctonias, which are a polyphyletic group of fungi belonging to Tulasnellaceae, Ceratobasidiaceae, and Serendipitaceae [3,41]. In this study, we collected substrates (e.g., litters, barks and mosses) from six original habitats of *D. officinale* and conducted ex situ seedling baiting experiments by transplanting in vitro-produced seedlings of *D. officinale* into the substrates. After cultivation for 75 days, various fungi colonized the roots of *D. officinale* seedlings and formed large numbers of pelotons in all six treatments. A total of 16 OMFs and 11 non-OMFs were finally isolated and identified. The 16 OMFs were typical OMFs, including 13 *Tulasnella* species and 3 species of Serendipitaceae.

Orchids recruit OMFs from ancestors colonizing roots as endophytes, and root endophytism in the orchid family was a predisposition for mycorrhizal evolution [42]. Rhizoctonias are known to live as saprobes in soil around the roots [3] or on tree bark around epiphytic orchids [43]. Therefore, the variation in mycorrhizal associations in orchid species is, to some extent, affected by specific environmental conditions [3,44] and the specificity of orchid species. For the ex situ seedling baiting developed in this study, several procedures for isolating the fungi, collecting materials, mixing, potting and orchid cultivation could affect the range of the fungi isolated. However, the dominant OMFs obtained in this study with high isolation frequencies (more than 10 times) were commonly isolated from more than four original habitats. Some studies suggest that orchids have a core set of keystone OMFs that are ubiquitously distributed and temporally stable, whereas the majority of OMFs are randomly associated with the plants [44,45]. The fungi associated with seedlings of *D. officinale* obtained in this study were different to those fungi that were isolated from protocorms in previous studies (Appendix A), and the dominant OMFs (*Tulasnella* TP-2, TP-8, TP-9, TP-11 and TP-13) were stably associated with seedlings of *D. officinale* across different habitats (Table 2), suggesting that those fungi were the keystone OMFs at the seedling stage.

### 4.2. In Situ vs. Ex Situ Seedling Baiting

Our understanding of the diversity and community composition of OMFs has increased greatly with the application of DNA sequencing technologies during the last three decades [46]. However, for most studies, the roots of adult orchid plants were sampled as study materials, and therefore, the results were strongly driven by differences in sampling efforts [44]. The diversity and species composition of OMFs may change among plant developmental stages [6,7,8]; for example, the mycoheterotrophic orchid *Gastrodia elata* uses *Mycena* for seed germination and *Armillaria mellea* for growth [47,48]. For most orchids, the differences in mycorrhizal associations at different developmental stages (e.g., the seed germination, seedling growth or flowering stage) remain largely unknown.

In our previous study, from naturally occurring protocorms, we successfully obtained effective fungi for seed germination and developed a method of using fungus-seed bags as propagules for the restoration-friendly cultivation of *D. officinale* [29]. To obtain OMFs supporting seedling growth, we further developed the method of in situ seedling baiting by transplanting in vitro-produced seedlings of *D. officinale* into their original habitats to capture fungi at the seedling stage. After one year, newly established roots were sampled six times during one year, and five *Tulasnella* species and one *Fusarium* species were obtained and identified [19]. Three *Tulasnella* isolates showed different impacts on seedling growth, despite their close phylogenetic relatedness. Finally, one *Tulasnella* isolate (*Tulasnella* sp. TPYD-2) showed a strong ability to promote seedling growth [19]. Compared to this method, the ex situ seedling baiting method used in the current study was more efficient at obtaining more OMFs in shorter time periods. Based on the results of the current study, we suggest that ex situ seedling baiting is presumably not only a relevant technique for over-collected orchids, but also presumably works for other orchids to obtain seedling-associated OMFs.

### 4.3. Fungi Associated with Dendrobium Officinale

The most common OMFs associated with *D. officinale* are species of Tulasnellaceae, and 25 out of 38 OMFs reported in 10 studies are from Tulasnellaceae (Appendix A). Species of *Tulasnella* are among the most common symbiotic fungi in orchids, and *Epulorhiza* is considered the counterpart of *Tulasnella* at the teleomorph stage [49]. In the current study, 13 out of 16 OMFs were *Tulasnella* species, including the top 5 OMFs with high isolation frequencies (more than 10 times; Table 2). Among the other OMFs associated with *D. officinale*, eight species of Sebacinales and three strains of *Mycena dendrobii* were reported in seven previous studies (Appendix A). In this study, three fungi of Serendipitaceae were obtained, and TP-15 was commonly isolated nine times from five sources (Table 2).

Among the 16 OMFs obtained in the current study, in addition to the three species of Serendipitaceae in Clade I of the Sebacinales tree (Figure 3a), three fungi of *Tulasnella* (TP-6, TP-8 and TP-13) were clustered in Clade II of the Tulasnellaceae tree with four other fungi of *Tulasnella* (TPYD-1, TPYD-2, TPYD-3 and TPYD-5) that were isolated from seedlings of *D. officinale* via in situ seedling baiting in our previous study [19], seven fungi of *Tulasnella* were clustered in Clade III with four other OMFs, and only three fungi (TP-1, TP-3 and TP-4) with relatively low isolation frequencies were included in Clade IV and Clade V with six fungi/strains obtained from adult roots or protocorms of *D. officinale* (Figure 3b). It is worth noting that the fungi obtained in the current study may not represent the whole OMF community, and procedures to isolate the fungi could affect the fungi isolated. The main aim of current study was to obtain cultivable OMFs associated with seedlings of *D. officinale*. To determine the OMF community composition at the seedling stage by transplanting seedlings into different habitats after the fungi are well-established in the seedling roots, Illumina MiSeq sequencing of the internal transcribed spacer 2 (ITS2) region should be applied [45].

In this study, 11 non-OMFs were also obtained and identified, and among them, *Fusarium oxysporum* TP-18 and *Muscodor* sp. TP-26 were the dominant endophytes, isolated 27 and 24 times, respectively. *F. oxysporum* was also isolated at a high frequency from seedlings of *D. officinale* via in situ seedling baiting in our previous study [19]. Various root-associated nonmycorrhizal endophytes were reported from many orchid species [50,51]. Among them, *F**. oxysporum* is a common fungus reported to be associated with various orchid species [28,52] and considered as a pathogen of many orchid species [53]. For *Dendrobium* species, *F. oxysporum* was reported to cause wilt disease in *D. officinale* [54], while other studies suggested that *Fusarium* species can enhance resistance to pathogens and promote plant growth in different species [55,56]. Many studies suggested that the border between endophytic and mycorrhizal fungi could be difficult to define, and some fungus–plant interactions can easily shift from mutualism to parasitism depending on the plant’s physiology and environmental conditions [57,58,59]. As discussed in our previous study, *F. oxysporum* was commonly associated with seedlings of *D. officinale*, and its effects alone, as well as its possible synergistic effects with other OMFs, on seedling growth are worth exploring [19].

## 5. Conclusions

To obtain plant growth-promoting fungi for orchid conservation, the traditional method is to isolate and screen OMFs from the roots of wild mature plants. However, because it is uncertain whether the fungi involved at the seedling stage remain until plant adulthood, this method might not always obtain the optimal source of fungal mycobionts; moreover, for many over-collected orchids, it is almost impossible to obtain wild plant materials. In this study, ex situ seedling baiting was developed to capture seedling-associated fungi in *D. officinale*. Finally, a total of 16 OMFs and 11 non-OMFs were successfully isolated. The OMFs were identified as different OTUs to fungi that were obtained from the protocorms or adult roots of *D. officinale* in previous studies, and the dominant OMFs were stably associated with seedlings of *D. officinale* across different habitats; this suggests that plants of *D. officinale* are associated with new fungal partners at the seedling stage. The functions of those OMFs on supporting seedling growth still need to be investigated in further studies. Our results suggest that this approach is an efficient and easy way to obtain seedling-associated fungi for orchids, especially over-collected orchids for which wild plants have disappeared or are rare in the field but the original habitats are known. This method could have broad applications in OMF studies, as well as in orchid conservation.

## Figures and Tables

**Figure 1 jof-08-01036-f001:**
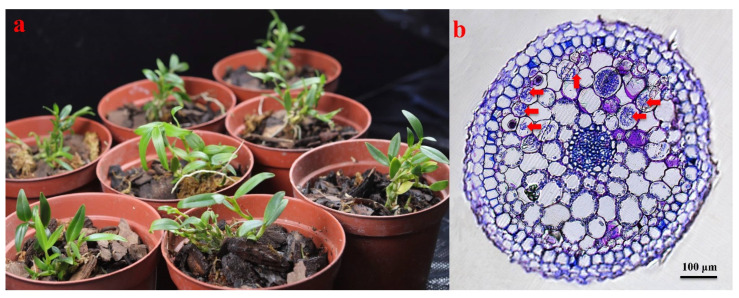
(**a**) The ex situ seedling baiting experiments conducted in this study, in which in vitro-produced seedlings of *Dendrobium officinale* were potted with substrates (e.g., litters, barks and mosses) collected from 6 original habitats of *D. officinale.* (**b**) Cross-section of *D. officinale* seedling root at 75 days after transplantation showing root colonization by orchid mycorrhizal fungi in DXLM treatment (red arrows indicating pelotons).

**Figure 2 jof-08-01036-f002:**
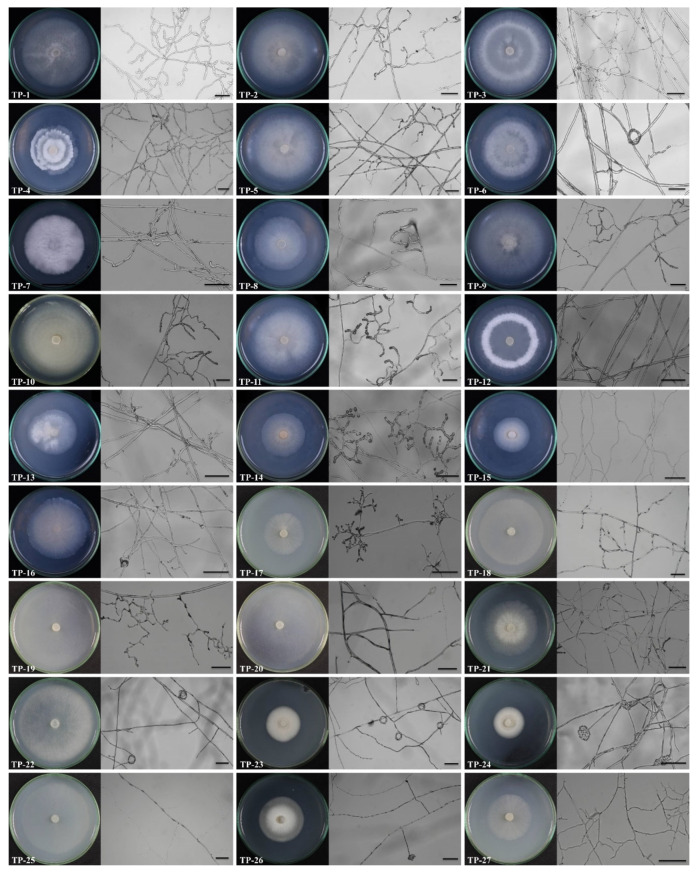
The colonies and in vitro morphological characteristics of hyphae for 27 fungi obtained in the current study 10 days after culture on PDA medium. The scale is 50 μm.

**Figure 3 jof-08-01036-f003:**
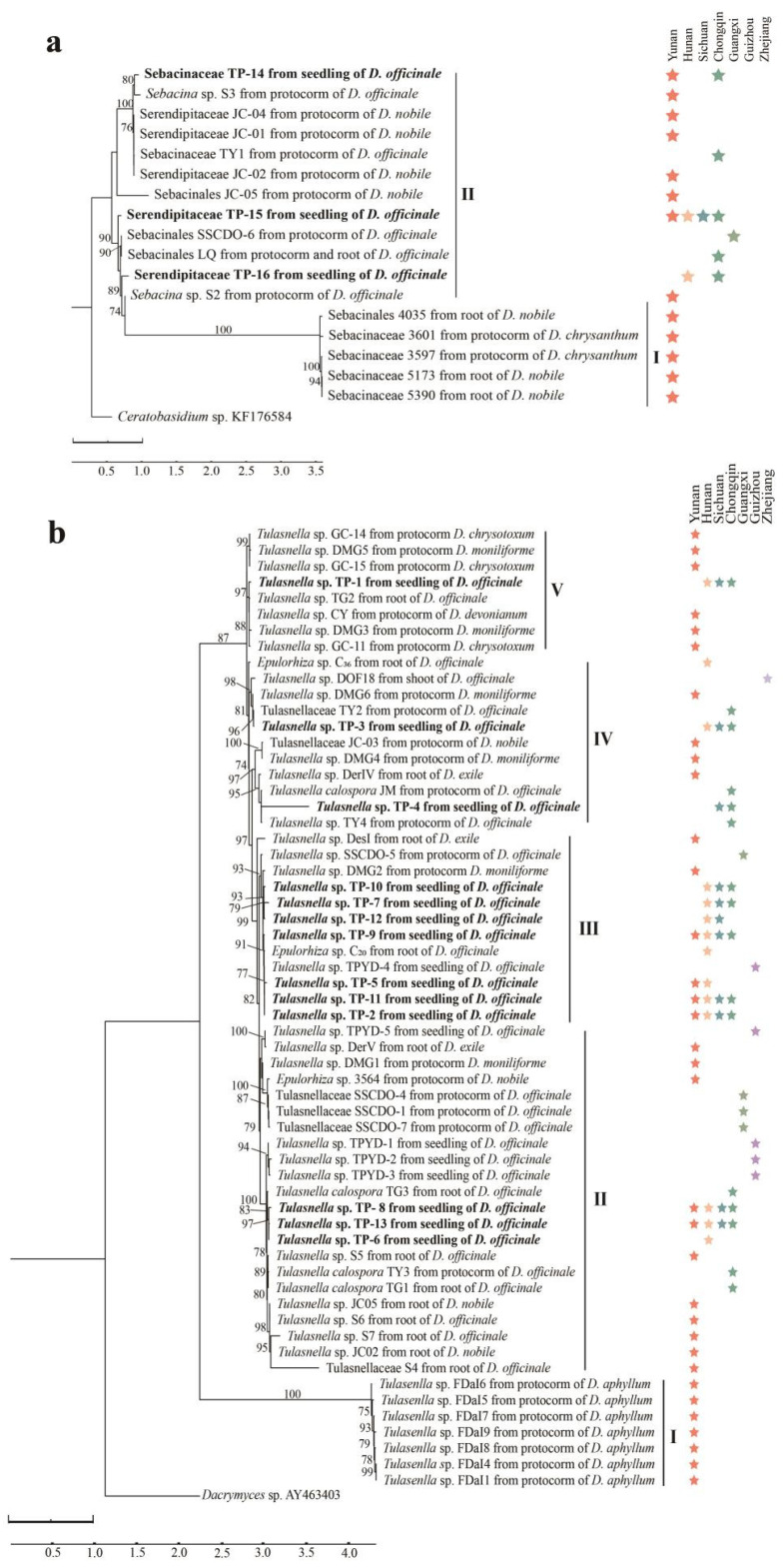
The two maximum-likelihood trees constructed using ITS sequences of 16 orchid mycorrhizal fungi (in bold) obtained from seedling roots of *Dendrobium officinale* via ex situ seedling baiting in this study and of other mycorrhizal fungi reported to be associated with *Dendrobium* species. (**a**) Maximum-likelihood tree for Sebacinales; (**b**) maximum-likelihood tree for Tulasnellaceae. Numbers above branches are bootstrap probabilities (whenever ≥70%) based on 1000 replicates.

**Table 1 jof-08-01036-t001:** The locations of six sampling sites with information about the growing habitats of *Dendrobium officinale* plants, materials collected for ex situ seedling baiting experiments and sampling dates.

Sampling Sites	Locations	Growing Habitats	Collected Materials	Sampling Date
**GN**, Guangnan County, Yunnan	23°58′ N, 105°11′ E; 1428 m alt.	On trees	Barks and mosses	3 August 2018
**DXLM**, Langshan Mountain, Hunan	26°20′ N, 110°46′ E; 455 m alt.	On sandy conglomerate of Danxia landform	Sandy conglomerate and mosses	11 August 2018
**KSLM**, Langshan Mountain, Hunan	26°30′ N, 111°10′ E; 340 m alt.	On rocks of karst landform	Mosses and litters	12 August 2018
**LD**, Luding County, Sichuan	29°23′ N, 102°21′ E; 1382 m alt.	On rocks	Mosses and litters	23 August 2018
**SM**, Shimian County, Sichuan	29°22′ N, 105°11′ E; 3596 m alt.	On rocks	Mosses and litters	24 August 2018
**LT**, Luotian Town, Chongqing	30°31′ N, 108°33′ E; 1200 m alt.	On rocks	Mosses and litters	25 August 2018

**Table 2 jof-08-01036-t002:** The fungi obtained via ex situ seedling baiting experiments in *Dendrobium officinale*, including 16 orchid mycorrhizal fungi (OMFs; TP1-TP16) and 11 non-OMFs (TP17-TP27) with their isolation frequencies from six original habitats.

Fungus Codes	Fungal Species	GenBank Accession Number	Sequence Ident. (%)	Closest Relative	Isolation Frequencies from Six Original Sources						
					GN	DXLM	KSLM	LD	SM	LT	Total
TP-1	*Tulasnella* sp.	MN918475	99	KM226996.1			3	1		1	5
TP-2	*Tulasnella* sp.	MN918476	99	GQ241863.1	4	3	2	2		3	14
TP-3	*Tulasnella* sp.	MN918477	99	KM211335.1			2		4	2	8
TP-4	*Tulasnella* sp.	MN918478	97	KX587480.1				1		1	2
TP-5	*Tulasnella* sp.	MN918479	99	GQ241817.1	5		2				7
TP-6	*Tulasnella* sp.	MN918480	99	KC291619.1		4					4
TP-7	*Tulasnella* sp.	MN918481	100	AB506862.1		2		2	1	1	6
TP-8	*Tulasnella* sp.	MN918482	98	KX587486.1	1	4			7	3	15
TP-9	*Tulasnella* sp.	MN918483	99	JX545220.1	1	1	6		3	8	19
TP-10	*Tulasnella* sp.	MN918484	99	FJ594913.1		2		1		5	8
TP-11	*Tulasnella* sp.	MN918485	99	HM214462.1	5	2	2		1	1	11
TP-12	*Tulasnella* sp.	MN918486	99	FJ594926.1		4		1			5
TP-13	*Tulasnella* sp.	MN918487	97	JX546238.1	3	3	4	4	4	2	20
TP-14	Species of Serendipitaceae	MN918488	99	EU668272.1	4					1	5
TP-15	Species of Serendipitaceae	MN918489	97	FJ788824.1	3	1		1	2	2	9
TP-16	Species of Serendipitaceae	MN918490	97	JX317218.1			1			1	2
	**The total number of OMFs/isolation times**				**8/26**	**10/26**	**8/22**	**8/13**	**7/22**	**13/31**	
TP-17	*Thanatephorus* sp.	MN918491	84	MH348617.1	2	4	1				7
TP-18	*Fusarium oxysporum*	MN918492	99	KU527806.1	2	7	17			1	27
TP-19	*Clitopilus* sp.	MN918493	96	KC176292.1				3	1		4
TP-20	*Plectosphaerella niemeijerarum*	MN918494	99	MG386080.1		1	1		2	6	10
TP-21	*Plectosphaerella cucumerina*	MN918495	99	MH673607.1			2	1	6	4	13
TP-22	Species of Ascomycota	MN918496	97	KT581843.1				2			2
TP-23	*Trichoderma* sp.	MN918497	100	MN602858.1	2		2			1	5
TP-24	*Trichoderma* sp.	MN918498	100	MN602619.1	1		2		1	1	5
TP-25	*Muscodor* sp.	MN918499	99	MK757898.1					6		6
TP-26	*Muscodor* sp.	MN918500	100	JN426991.1	2	2	7	7	3	3	24
TP-27	*Muscodor* sp.	MN918501	99	KM514680.1	1		1	4		1	7

## Data Availability

Publicly available datasets were analyzed in this study. These data can be found here: https://www.ncbi.nlm.nih.gov/, accessed on 16 December 2021.

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
