# Peer review of "Using Ex Situ Seedling Baiting to Capture Seedling-Associated Mycorrhizal Fungi in Medicinal Orchid Dendrobium officinale"

_jof, 2022, doi:10.3390/jof8101036_

Round 1

Reviewer 1 Report (Previous Reviewer 1)

Authors have made all improvements or recommendations for the manuscript.

Author Response

Responses to Reviewer 1:

Authors have made all improvements or recommendations for the manuscript.

R: Thanks for very valuable and constructive comments and suggestions.

Reviewer 2 Report (Previous Reviewer 2)

The authors have addressed almost all my previous concerns in this revised version of the manuscript. However, I found a few errors in the new text that should be taken care of before submitting a final version for publication.

Minor issues

L61: The authors wrote, "we used this method ...," but I wonder if the "this" shows in situ or ex situ bating? Or both? Please clarify what you mean.

L149: Add a space at "cells.Once..."  

Author Response

Responses to Reviewer 2:

The authors have addressed almost all my previous concerns in this revised version of the manuscript. However, I found a few errors in the new text that should be taken care of before submitting a final version for publication.
R: Thanks for very valuable and constructive comments and suggestions.

Minor issues

L61: The authors wrote, "we used this method ...," but I wonder if the "this" shows in situ or ex situ bating? Or both? Please clarify what you mean.

R: Here it refers to both in situ and ex situ seed bating, and I now have changed the sentence as : we used in situ/ex situ seed baiting to ...

   The references were also cited here [24-29], e.g., in situ seed baiting for Dendrobium aphyllum [24], Dendrobium devonianum [25], Dendrobium moniliforme [27], Dendrobium exile [28], Dendrobium officinale [29] and ex situ seed baiting for Arundina graminifolia [26].

L149: Add a space at "cells.Once..."  

R: Changed.

This manuscript is a resubmission of an earlier submission. The following is a list of the peer review reports and author responses from that submission.

Round 1

Reviewer 1 Report

line 12 "substances" change by substrates

line 13 "in different geografic locations" add country

lines 14-15 "After ... ... six groups" clarify wording

lines 17-18

line 23 replace "for orchids" by "for this species" and could be performed for other overcollected species 

lines 30-39 the arguments are generalists and can be inmediately start for the importance of OMF in orchids. Dendrobium genera is specific to Tullasnellaceae and this must be argued this specificity in the orchid family.

63-66 The argument about the variations in the changing of mycorrhizal partners in seeds and seedlings in plant performance are not discussed as the justification for this research

line 68 please verify drafting

78-79, please say a line about the importance of the "well-studied" species or at least put the main references.

110 Please, specify the way to make the mix for each locality substrate when materials are so heterogeneous. How you assured a wood homogeneization? How many points was taken for the sampled mixture?

line 112-113 specify the sterilization way of this substrates because they are a strong source of fungi associated with. The mixtures are V:V.V?

117 please specify the seedling stage or criteria after in vitro propagation: how many time after germination. Have used1800 seedlings ?

127 the cross-section is only for one treatment, specify and add. "aspect of root colonization by orchid mycorrhizal fungi in ...."

between line 131-132 in Table correct the orthography

line 136 please argue the mean of "very large"? it is a high colonization percentage? The hyphal coils are undigested as synonim of very large? the time of this was the same for all treatments? or was between a period of 30 to 90 days?

line 142 unnecesary to specify the brand

line 143, please specify the pH

line 150, please specify the mean of colours by a way to selection.

line 153 here it is important to specify the DNA extraction kit

line 162 and Table S1, have selected only the OMF associated wit Dendrobium or isolated from Dendrobium??

line 181 please replace by habitats

line 193 delete dot.

line 213 please orthography

line 227 The growth of the colonies are in the same time?

line 242, please separate

line 253, 255 and 355, please substrates

263-264 The variation in mycorrhizal associations isd also due to specificity as is  extensively documented in this genus

269 sources in this case are related to habitats of orchid so, please remain with the same sense.

269-271 it is unnecesary to repeat the same in eg. It will be better to argue analyze better the differences between protocorm and seedling OMF communities by site from the previous study.

line 273 in this sense, ubiquity could be discussed as the related fungi in the Table 1 for the gene bank identity. All species related with this has been found where?

lines 301-306 I put this paragraph in introduction

lines 308-309. If some the studies in STable are from sequencing directly from root segments, OMF are more a specificity phenomenon than "most common OMF associated" for the Sebacinales. 

lines 320-21 It will be better to argue the question in lines 63-66 that has not been addressed in relation to heterogeneity in the development of plants related to structure of OMF

lines 354-359. Conclusions is not about repeat data as in abstract, you can highlight other aspects of prvious studys in OMF changes or affinities in taxonomic OTUS

Author Response

Responses to Reviewer 1:

line 12 "substances" change by substrates  

R: The "substances" have been replaced by “substrates” throughout the text.

line 13 "in different geografic locations" add country

R: Done.

lines 14-15 "After ... ... six groups" clarify wording

R: The sentence now has been changed as After cultivation for 75 days, it was obviously observed that fungi colonized seedling roots and formed large numbers of pelotons in all six groups.

lines 17-18

R: The sentence now has been changed as “The fungal species isolated from the different groups (original habitat sources) were not identical....

line 23 replace "for orchids" by "for this species" and could be performed for other overcollected species 

R: Changed as suggested.

lines 30-39 the arguments are generalists and can be inmediately start for the importance of OMF in orchids. Dendrobium genera is specific to Tullasnellaceae and this must be argued this specificity in the orchid family.

R: This paragraph is a brief introduction of plant mycorrhiza, enabling the general readers to also understand the background of current study and the special field of orchid mycorrhiza. So, I would like to keep the paragraph here. Thanks!

The current knowledge about OMFs do not support that “Dendrobium genera is specific to Tullasnellaceae”, and fungi from Mycena, Sebacinaceae, Ceratobasidiaceae and Serendipitaceae have also been reported associating with Dendrobium species (See Appendix Table S1 of current submission).

63-66 The argument about the variations in the changing of mycorrhizal partners in seeds and seedlings in plant performance are not discussed as the justification for this research

R: I agree with reviewer, and now have deleted this paragraph.

line 68 please verify drafting

R: Changed.

78-79, please say a line about the importance of the "well-studied" species or at least put the main references.

R: The paragraph in Discussion about Dendrobium officinale (lines 301-306) now has been moved here as suggested by reviewer below.

110 Please, specify the way to make the mix for each locality substrate when materials are so heterogeneous. How you assured a wood homogeneization? How many points was taken for the sampled mixture?

R: In this study, samples were took from 6 original habitats of Dendrobiu officinale. In each site, the substrates were collected within 30 cm around a plant of Dendrobiu officinale. For GN site, plants of D. officinale grew on branches of tree, so we collected barks and mosses. At DXLM site, plants grew on sandy conglomerate and sandy conglomerate with mosses were collected. For other four sites, plants grew on rocks and mosses and litters on the surface of rocks were collected (see Table 1). For each site, available above materials were randomly collected for about 5 kg and put together as original substrates and mixed with sterilized mixed substrates after we returned laboratory.

To make it clear, I now have rewritten this sentence as: At each site, available materials within 30 cm around the plants were randomly collected for about 5 kg and put together as original substrates, and transferred to the laboratory within 48 h.

line 112-113 specify the sterilization way of this substrates because they are a strong source of fungi associated with. The mixtures are V:V.V?

R: In this study, bark, peat and volcanic stone were mixed in a ratio of 2:1:1 in volumes and then autoclaved at121℃ for 30mins. I now have changed this sentence to make it clear.

117 please specify the seedling stage or criteria after in vitro propagation: how many time after germination. Have used 1800 seedlings?

R: To make it clear, the sentence has been changed as: and then asymbiotically germinated on MS medium [34] for about 8 months to obtain in vitro-produced seedlings with 5-6 leaves for ex situ seedling baiting experiment.

Yes, a total of 1800 seedlings were used (5 seedlings * 60 pots * 6 treatments).

127 the cross-section is only for one treatment, specify and add. "aspect of root colonization by orchid mycorrhizal fungi in ...."

R: Changed as suggested.

between line 131-132 in Table correct the orthography

R: Corrected.

line 136 please argue the mean of "very large"? it is a high colonization percentage? The hyphal coils are undigested as synonim of very large? the time of this was the same for all treatments? or was between a period of 30 to 90 days?

R: Here, it means many pelotons could be obviously found in the roots. For all six treatments, at 75 days after cultivation, we observed many pelotons formed in the roots of D. officinale plants, and began to sample roots for fungal isolation. I now have changed very large pelotons as large numbers of pelotons”.

line 142 unnecesary to specify the brand

R: Deleted.

line 143, please specify the pH

R: Done.

line 150, please specify the mean of colours by a way to selection.

R: The morphological studies on fungal strains now have been described in detail.

line 153 here it is important to specify the DNA extraction kit

R: I now have added using Fungi Genomic DNA Extraction Kit (D2300, Solarbio).

line 162 and Table S1, have selected only the OMF associated with Dendrobium or isolated from Dendrobium??

R: The fungi listed in Table S1 were all isolated from Dendrobium species and culturable.

line 181 please replace by habitats

R: Changed as suggested.

line 193 delete dot.

R: Done.

line 213 please orthography

R: Corrected.

line 227 The growth of the colonies are in the same time?

R: Yes, all 27 fungi strains were cultured for 10 days on PDA medium. To make it clear, I now have added At 10 days after culture on PDA medium.

line 242, please separate

R: Done.

line 253, 255 and 355, please substrates

R: Changed.

263-264 The variation in mycorrhizal associations is also due to specificity as is extensively documented in this genus

R: I agree, and now have added and specificity of orchid species at the end of this sentence.

269 sources in this case are related to habitats of orchid so, please remain with the same sense.

R: Agree, and I now have replaced sources by habitats.

269-271 it is unnecesary to repeat the same in eg. It will be better to argue analyze better the differences between protocorm and seedling OMF communities by site from the previous study. line 273 in this sense, ubiquity could be discussed as the related fungi in the Table 1 for the gene bank identity. All species related with this has been found where?

R: I agree with reviewer, and tried my best to reorganize this paragraph. Hope it could make sense now. It now goes as: However, the dominant OMFs obtained in this study with high isolation frequencies (more than 10 times) were commonly isolated from more than 4 original habitats. Some studies suggested that orchids have a core set of keystone OMFs that are ubiquitously distributed and temporally stable, whereas the majority of OMFs are randomly associated with the plants [43,44]. The fungi associated with seedlings of D. officinale  obtained in this study were different to those fungi that were associated with protocorms in previous studies (Table S1) , and the dominant OMFs (Tulasnella TP-2, TP-8, TP-9, TP-11 and TP-13) stably associated with seedlings of D. officinale across different habitats (Table 2), suggesting those fungi were the keystone OMFs at seedling stage.  

lines 301-306 I put this paragraph in introduction

R: Changed as suggested.

lines 308-309. If some the studies in STable are from sequencing directly from root segments, OMF are more a specificity phenomenon than "most common OMF associated" for the Sebacinales. 

R: The fungi listed in Table S1 were all isolated from Dendrobium species and culturable.

lines 320-21 It will be better to argue the question in lines 63-66 that has not been addressed in relation to heterogeneity in the development of plants related to structure of OMF

R: As previously suggested by the reviewers, the paragraph of the question in lines 63-66 has been deleted. I agree with reviewer it is hard to address or discuss the question in relation to heterogeneity in the development of plants from the results of current study.

lines 354-359. Conclusions is not about repeat data as in abstract, you can highlight other aspects of previous studies in OMF changes or affinities in taxonomic OTUS

R: Thanks for the suggestions. The paragraph now has been rewritten as: In this study, ex situ seedling baiting was developed to capture seedling associating fungi in D. officinale. Finally, a total of 16 OMFs and 11 non-OMFs were successfully isolated. The OMFs were identified as different OTUs to fungi that were obtained from protocorms or adult roots of D. officinale in previous studies, and the dominant OMFs stably associated with seedlings of D. officinale across different habitats, suggesting that plants of D. officinale associated with new fungal partners at seedling stage. The functions of those OMFs on supporting seedling growth still need to be investigated in further studies.

Reviewer 2 Report

Major comments

Wu et al reported the ex-situ seedling bating method using substances collected from original habitats of Dendrobium officinale to obtain seedling-associated fungi. Several fungi, including the common OMFs, were isolated from the roots of plants grown for a period of time. The authors concluded that the ex-situ seedling bating has a great potential to isolate beneficial fungi for epiphytic orchids. The manuscript was well written, but my main concern is that the seedlings were used for the bating. There is no data on whether the isolated fungi can contribute to seed germination. The authors described that the OMFs isolated from adult roots are not always support their growth or development because of the symbiont switching depending on the orchid life stages. This is really a problem in the case of switching the symbionts between seed germination and plant growth. In fact, many orchids cannot germinate in vitro. I think the authors should test the ex-situ method using seeds rather than seedlings or verify the contribution to seed germination using isolated fungi. Also, I wonder if the ex-situ approach is limited to epiphytic orchids or can apply to terrestrial orchids. What does the author think about this?

My second concern is that there was no data on the contribution of the isolated fungi to the plant. Which fungus is the best contributor? The author should quantify the plant growth.

Minor comments

1. In the introduction, add the information on whether D. officinale switches their symbionts depending on the life stages.

2. L110: The authors described "collected materials were mixed together," but I'm not sure in what proportion the authors mixed them.

3. Describe the size of the pot.

4. How far did the authors take the substances from where the plants grew? Describe the sampling methods clearly.

5. How big were the seedlings, and how many days did the authors transplant them after growing them?

6. L135: Describe the method of the trypan blue staining briefly.

7. L139-140: Describe the method of root surface sterilization.

8. L153-155: The authors cited literature. However, briefly describe the method of DNA extraction and PCR.

9. Add the explanation about the 30, 45, 60, and 90 days cultivated plant. Were these plants inadequate for isolating fungi?

10. Fig. 3: Use a nearer outgroup for making an ML tree of Sebacina isolates. Try something from Cantharellales or Auriculariales rather than Agaricales. I will suggest even Ceratobasidium or Paulisebacina. Armillaria is too far. And if possible, try to show the real distance instead of topology. For Tulasnella tree, I agree with Dacrymyces, but I am curious about the real distance.

Author Response

Responses to Reviewer 2:

Wu et al reported the ex-situ seedling bating method using substances collected from original habitats of Dendrobium officinale to obtain seedling-associated fungi. Several fungi, including the common OMFs, were isolated from the roots of plants grown for a period of time. The authors concluded that the ex-situ seedling bating has a great potential to isolate beneficial fungi for epiphytic orchids. The manuscript was well written, but my main concern is that the seedlings were used for the bating. There is no data on whether the isolated fungi can contribute to seed germination. The authors described that the OMFs isolated from adult roots are not always support their growth or development because of the symbiont switching depending on the orchid life stages. This is really a problem in the case of switching the symbionts between seed germination and plant growth. In fact, many orchids cannot germinate in vitro. I think the authors should test the ex-situ method using seeds rather than seedlings or verify the contribution to seed germination using isolated fungi. Also, I wonder if the ex-situ approach is limited to epiphytic orchids or can apply to terrestrial orchids. What does the author think about this? My second concern is that there was no data on the contribution of the isolated fungi to the plant. Which fungus is the best contributor? The author should quantify the plant growth.

R: Thanks for very valuable and constructive comments and suggestions.

In our previous studies, some OMFs were isolated from naturally occurred protocorms of Dendrobium officinale. Among them, fungus Sebacinales LQ could quickly promote seed germination up to seedling stage, and has been successfully used in conservation practices of D. officinale (Wang et al., 2021. Isolating ecological-specifc fungi and creating fungus-seed bags for epiphytic orchid conservation. Glob. Ecol. Conserv. 2021, 28, e01714). However, we found that after seeds germinated into seedlings, seedling growth showed great variations even in a small-scale environment. So, we were thinking if switching of the symbionts between seed germination and plant growth occurred. In current study, we aimed to develop the method of using ex situ seedling baiting to capture seedling associating OMFs. As reviewer pointed out that it is important to test the contribution of the isolated fungi to the plant, and we will conduct experiments to test the functions of OMFs on seed germination and seedling growth, and further screen out seedling growth-promoting fungi.

To obtain orchid seedling associating fungi for further screening of seedling growth-promoting fungi, the traditional way is to isolate fungi from roots of wild adult plants, which has been reported in many previous studies. However, this method has many defects in practice, e.g., hard to collect root samples, difficult to isolate and screen optimal source of fungi because a high diversity of fungi may associate with adult plant roots and orchids may associate with different mycorrhizal fungi at different life stages. In current study, by using ex situ seedling baiting, 16 OMFs associated with seedlings of Dendrobium officinale were obtained successfully. It would be much easy to screen seedling growth-promoting fungi from those OMFs associated with seedlings. So we think the approach of ex situ seedling baiting could also be used in other orchids including terrestrial orchids.

Minor comments

  1. In the introduction, add the information on whether D. officinale switches their symbionts depending on the life stages.

R: So far, there is no literature or studies clearly address this issue in D. officinale. Based on our previous studies, I now have added a paragraph as: In our previous studies, by transplanting in vitro-produced seedlings of D. officinale into their original habitats, we used newly established roots of well-growing seedlings to isolate fungi after more than one year, and successfully obtained five Tulasnella species, including one isolate being able to strongly promote seedling growth [19]. The fungus has not been reported among OMFs isolated from protocorms or adult roots of D. officinale in previous studies, suggesting that the symbiont switching may occur depending on life stages in D. officinale and seedlings of D. officinale established new fungal partners that in turn played an important role in their growth.

  1. L110: The authors described "collected materials were mixed together," but I'm not sure in what proportion the authors mixed them.

R: It was also concerned by reviewer 1. In this study, samples were took from 6 original habitats of Dendrobiu officinale. In each site, the substrates were collected within 30 cm around a plant of Dendrobiu officinale. For GN site, plants of D. officinale grew on branches of tree, so we collected barks and mosses. At DXLM site, plants grew on sandy conglomerate and sandy conglomerate with mosses were collected. For other four sites, plants grew on rocks and mosses and litters on the surface of rocks were collected (see Table 1). For each site, available above materials were randomly collected for about 5 kg and put together as original substrates and mixed with sterilized mixed substrates after we returned laboratory.

To make it clear, I now have rewritten this sentence as: At each site, available materials within 30 cm around the plants were randomly collected for about 5 kg and put together as original substrates, and transferred to the laboratory within 48 h.

  1. Describe the size of the pot.

R: It was the circular plastic pots with 8 cm in height and 10 cm in diameter. I now have make it clear in the text.

  1. How far did the authors take the substances from where the plants grew? Describe the sampling methods clearly.

R: At each site, available materials within 30 cm around the plants were randomly collected for about 5 kg and put together as original substrates. I now have described the sampling methods in detail in the text.

  1. How big were the seedlings, and how many days did the authors transplant them after growing them?

R: The seedlings have been described in detail as: and then asymbiotically germinated on MS medium [34] for about 8 months to obtain in vitro-produced seedlings with 5-6 leaves for ex situ seedling baiting experiment.

  1. L135: Describe the method of the trypan blue staining briefly.

R: The methods have been described now (Lines 146-151).

  1. L139-140: Describe the method of root surface sterilization.

R: Done as suggested.

  1. L153-155: The authors cited literature. However, briefly describe the method of DNA extraction and PCR.

R: Done as suggested (Lines 169-175).

  1. Add the explanation about the 30, 45, 60, and 90 days cultivated plant. Were these plants inadequate for isolating fungi?

R: To make sure when is the best time point to sample roots for fungi isolation, we experimental designed to sample roots from each group at 30, 45, 60, 75 and 90 days after cultivation, and stain and observe the presence of pelotons. In this study, the result was that at 75 days after cultivation large numbers of pelotons were observed in the roots in all six treatments, so we started to sample roots for fungal isolation. To make it clear, I now have added a sentence as:

To determine the time point of root sampling for fungi isolation, three roots from each group were randomly sampled and stained to examine for the presence of pelotons at 30, 45, 60, 75 and 90 days after cultivation, respectively.

  1. Fig. 3: Use a nearer outgroup for making an ML tree of Sebacina isolates. Try something from Cantharellales or Auriculariales rather than Agaricales. I will suggest even Ceratobasidium or Paulisebacina. Armillaria is too far. And if possible, try to show the real distance instead of topology. For Tulasnella tree, I agree with Dacrymyces, but I am curious about the real distance.

R: Thanks for the suggestion. We now have used Ceratobasidium sp. (KF176584) as outgroup sequences of Serendipitaceae, and re-generated phylogenetic trees and showed the real distances (Figure 3a,b).

Round 2

Reviewer 2 Report

After revising the manuscript, I feel that the authors have addressed all my raised points. I have no further comments or recommendations to make.